# Effect of Ionic Strength on Thioflavin-T Affinity to Amyloid Fibrils and Its Fluorescence Intensity

**DOI:** 10.3390/ijms21238916

**Published:** 2020-11-24

**Authors:** Kamile Mikalauskaite, Mantas Ziaunys, Tomas Sneideris, Vytautas Smirnovas

**Affiliations:** 1Institute of Biotechnology, Life Sciences Center, Vilnius University, LT-10257 Vilnius, Lithuania; kamile.mikalauskaite@gmail.com (K.M.); mantas.ziaunys@gmail.com (M.Z.); sneideris.t@gmail.com (T.S.); 2Department of Chemistry, University of Cambridge, Cambridge CB2 1EW, UK

**Keywords:** amyloid, protein aggregation, ionic strength, thioflavin-T

## Abstract

The formation of amyloid fibrils is linked to multiple neurodegenerative disorders, including Alzheimer’s and Parkinson’s disease. Despite years of research and countless studies on the topic of such aggregate formation, as well as their resulting structure, the current knowledge is still fairly limited. One of the main aspects prohibiting effective aggregation tracking is the environment’s effect on amyloid-specific dyes, namely thioflavin-T (ThT). Currently, there are only a few studies hinting at ionic strength being one of the factors that modulate the dye’s binding affinity and fluorescence intensity. In this work we explore this effect under a range of ionic strength conditions, using insulin, lysozyme, mouse prion protein, and α-synuclein fibrils. We show that ionic strength is an extremely important factor affecting both the binding affinity, as well as the fluorescence intensity of ThT.

## 1. Introduction

Protein aggregation into insoluble, highly structured amyloid fibrils is related to the onset and progression of many neurodegenerative disorders, such as Alzheimer’s or Parkinson’s diseases [1,2]. Despite an abundance of experiments conducted with both model [3,4] and disease-related proteins [5,6] there is still a limited understanding of how native proteins convert to these beta-sheet rich aggregates [7]. In addition, very few potential anti-amyloid compounds have passed the initial clinical trials, and none have been approved as effective in treating or curing patients [8,9]. These two factors are intertwined, as a limited comprehension of protein fibrillization and the methods used to track it ultimately led to the identification of seemingly potential, yet ineffective, disease-modulating compounds.

There are multiple methods used to track protein aggregation into amyloid fibrils. Changes in their secondary structure can be analyzed by circular dichroism [10] or Fourier-transform infrared spectroscopy [11]; aggregate morphology is commonly examined by transmission electron microscopy [12] or atomic force microscopy [13], while changes in fibril quantity are determined by sedimentation or amyloidophilic dye binding [14]. Despite the variety of methods, each one has its limitations, such as the inability to detect different types of aggregates or to quantify their concentration in solution. When examining a potential anti-amyloid compound, these drawbacks could lead to a false interpretation of the results and yield another failed clinical trial.

One of the more commonly used spectroscopic methods to track fibrillization reactions is a thioflavin-T (ThT) fluorescence assay [15]. This molecule binds to the beta-sheet grooves on the fibril’s surface and attains a locked conformation, resulting in a significant increase in its fluorescence emission intensity [16,17]. Changes in this intensity are used as an indicator of fibril assembly or disassembly [18]. If a decrease in signal intensity is observed, it is attributed to the reduction of amyloid assemblies [19,20], caused by the tested anti-amyloid compound. In recent years, such a correlation between the quantity of fibrils and ThT fluorescence intensity has come into question, as multiple reports displayed a variety of factors that can modulate this dye’s fluorescence potential [21,22,23]. It was shown that distinct fibril conformations, originating from the same protein, can possess different bound ThT fluorescence intensities [24,25,26]. Moreover, the signal intensity can be modulated by other compounds present in solution, either by fluorescence quenching, an inner filter effect or interactions on the fibril’s surface [22,27,28]. In addition, to complicate matters further, fibrils can bind ThT in more than one type of binding mode with different affinities [29,30]. The binding of ThT is also not limited to just amyloid fibrils, but to certain specific native state proteins [31].

One factor that was observed on a few occasions was that a change in the solution’s ionic strength modulated ThT binding to lysozyme, prion protein fragment and amyloid beta fibrils [32,33,34]. A higher salt concentration resulted in a significant change to the dye’s fluorescence intensity. Experiments with amyloid proteins are conducted in a wide variety of conditions and the addition of various salts, such as sodium chloride [10,35] or guanidine hydrochloride [36,37,38], are often used to initiate or speed up aggregation reactions. Because of this, the high variety of ionic strength conditions may also result in different types of ThT binding, making fluorescence intensity comparisons completely inaccurate and irrelevant.

In this work, we examine ThT binding to amyloid fibrils formed of either model amyloidogenic proteins–insulin [39] and lysozyme [3], or neurodegenerative disease-related prion protein [40] and α-synuclein [41] under a large range of ionic strength conditions. We compare the differences in total bound ThT concentration, its fluorescence intensity, self-quenching ability, and possible new binding modes and show that ionic strength has a significant effect on all of these factors.

## 2. Results

Before examining the interaction between amyloid fibrils and ThT under a range of NaCl concentrations, the effect of ionic strength was examined on the dye molecule itself. Based on the absorbance spectra and calculated extinction coefficient values at 412 nm, there does not seem to be any significant effect that NaCl has on non-bound ThT, even at the highest ionic strength conditions (Appendix A
Figure A1). In the case of such small variations having any effect, the determined condition-specific ε_412_ values were used in all subsequent calculations.

When insulin fibrils are sonicated and resuspended into a range of NaCl and ThT concentration solutions, the first relevant observation is a difference in sample optical density at 600 nm (Figure 1A). When there is no NaCl present in solution, the optical density (OD_600_) is relatively low (0.1). It then increases with the addition of NaCl to roughly 0.35 and reaches a plateau. This suggests that up to a certain ionic strength, insulin fibrils are less prone toward self-association. This is further supported by subsequent sample centrifugation, where a substantial concentration of residual fibrils was still present in the supernatant at low ionic strength conditions (0–10 mM NaCl). This has, in turn, made it difficult to accurately determine the concentration of free and bound ThT molecules for solutions containing 0 and 10 mM NaCl (Figure 1B).

Following centrifugation, the concentration of free and bound ThT was determined for samples containing a range of NaCl (from 20 mM to 2 M) and ThT (from 20 µM to 100 µM) concentrations (Figure 1B). There is a direct correlation between the concentration of bound ThT and the solution’s ionic strength, as well as total ThT concentration. The dye-binding capacity of insulin fibrils increases from 4 µM at lower ionic strength and ThT concentrations, up to 55 µM at the highest NaCl and dye concentrations. The effect of ionic strength is most evident at the highest ThT concentration, where we observe a 4-fold increase in bound molecule concentration upon increase of NaCl concentration from 20 mM to 2 M (Figure 2B).

The fluorescence intensity follows a similar tendency, with higher ionic strength and ThT concentrations yielding a stronger signal (Figure 1C). However, the maximum fluorescence intensity values are not located at the same position as the highest concentration of bound dye, but rather at the intermediate level. This suggests that there is a critical concentration of bound ThT molecules, after which the self-quenching effect [42] overcomes the increase in fluorescence-capable dye molecules. Dividing the signal intensity by the concentration of bound ThT reveals that the fluorescence quantum yield decreases in an arc shape (Figure 1D), suggesting that the more molecules are bound to the fibril’s surface, the more they experience fluorescence self-quenching. Interestingly, the quantum yield values overlap with one another between different NaCl concentration conditions (Figure 1D), indicating that ionic strength itself does not influence the fluorescence intensity, but only the concentration of bound ThT molecules. 

When examining the ionic strength’s effect on lysozyme fibrils (Figure 2A), similar tendencies are observed. When there is no NaCl present in solution, its OD_600_ is relatively low and, as in the case of insulin fibrils (Figure 1A), it rapidly rises with the increasing NaCl concentration, subsequently reaching a plateau. This change, however, appears to be more extreme than in the case of insulin fibrils, as the OD_600_ value goes from less than 0.1 to 0.5–0.6, while in the former case it was from 0.1 to 0.35. This suggests that lysozyme fibril self-association is more sensitive to changes in the solution’s ionic strength. The plateau is also reached at a slightly lower solution’s ionic strength than in the case of insulin fibrils.

While the dependence between bound ThT and ionic strength/total ThT concentration pertains a similar tendency (Figure 2B) as with insulin fibrils, the concentration of bound dye at the maximum point is considerably higher (~70 µM as opposed to ~50 µM). The maximum fluorescence values are also shifted toward higher NaCl and ThT concentrations. The higher bound dye molecule concentrations result in a lower quantum yield, which shifts from 30 µM*^−^*^1^ to 10 µM*^−^*^1^ (Figure 2D), as opposed to insulin’s 50 µM*^−^*^1^ to 10 µM*^−^*^1^ (Figure 1D) due to an increase in fluorescence self-quenching. In this case, there is a less precise overlap between fluorescence quantum yield values at different NaCl concentrations; however, they still follow a similar arc shape as with insulin fibrils.

Contrary to insulin and lysozyme, MoPrP fibrils appear to require a relatively high ionic strength (100 mM NaCl) to begin an effective self-association (Figure 3A). This makes it quite difficult to accurately determine the concentrations of bound ThT molecules (Figure 3B); however, there still seems to be a similar binding tendency as with both other protein fibrils, but with a lower maximum bound ThT concentration.

The fluorescence intensity value distribution also has a unique pattern, with intensity values being low up to 50–100 mM NaCl, after which the signal values suddenly increase (Figure 3C). This distribution is similar to the OD_600_ values (Figure 3A), indicating that there may be a correlation between fibril self-association tendencies and the maximum fluorescence intensity. The quantum yield values follow the same arc shape as in both other cases.

The most interesting effect of ionic strength appears in the case of α-synuclein fibrils. While the changes in OD_600_ (Figure 4A) and bound ThT concentration (Figure 4B) share similar tendencies to MoPrP fibrils, the fluorescence intensity itself is much higher than in all other three cases (Figure 4C), while OD_600_ is the lowest out of all four protein fibrils. This indicates that the fibrils are, in general, considerably less self-associated and bound ThT molecules have a significantly higher fluorescence quantum yield. Another interesting aspect is that the effect of ThT self-quenching appears to be much greater, as the fluorescence intensity decreases with increasing dye concentration even at the lowest NaCl concentrations (Figure 4C), opposite to what was observed for other protein fibrils.

An unusual distribution in ThT quantum yield values is also observed (Figure 4D). In the presence of 100 mM or 200 mM NaCl, the I/c_B_ values do not overlap with all the others, which were determined at higher ionic strength conditions. This indicates the possibility of a significant change in either ThT binding or the fibrils themselves. In fact, when all four aggregates were examined using Fourier-transform infrared spectroscopy (FTIR), the secondary structure of α-synuclein fibrils was affected by the change in NaCl concentration, while there were no substantial differences observed in the case of insulin, lysozyme of MoPrP fibrils (Appendix A
Figure A2). The FTIR signal intensity at 1622 cm*^−^*^1^ (associated with beta-sheets) [43] is higher in the presence of 2 M NaCl. At low ionic strength, the aggregates appear to have a substantially larger disordered part, when compared to fibrils at high ionic strength. This shift in FTIR spectra and the ThT quantum yield values suggests an ionic strength-induced conformational change.

The excitation-emission matrix (EEM) maximum signal intensity at low bound ThT concentrations is located at different excitation-emission wavelengths for all four types of fibrils (Figure 5A–D). This is to be expected, as distinct fibrils can possess specific ThT binding modes [17]. As the concentration of bound dye increases, the insulin, lysozyme and MoPrP EEM position shifts toward lower excitation and higher emission wavelengths, while in the case of α-synuclein, it experiences a very minor change toward a higher excitation wavelength. In general, such an observation can have two different explanations. One possibility is that an increase in the solution’s ionic strength can result in ThT binding in a different mode on the fibril’s surface, which has a specific maximum excitation/emission wavelength. The second viable explanation is that these amyloid fibrils have multiple binding modes with specific capacities. If the increase in the solution’s ionic strength enhances ThT-fibril association, then certain binding modes may reach their capacity limit, causing more dye molecules to bind in a different mode and, in turn, result in a shift of the EEM maximum position. In the case of α-synuclein fibrils, a third possibility exists, where the change may be related to there being two different fibril conformations at both ends of the ionic strength spectrum.

## 3. Discussion

One of the more interesting events observed in this work is the apparent loss or reduction of fibril self-association properties at low ionic strength conditions. This could be attributed to being a generic feature of amyloid fibrils; however, both the change in OD_600_ values and the NaCl concentration at which the change occurs differ for all four protein fibrils. Lysozyme fibrils require the lowest ionic strength to self-associate, with insulin fibrils following a close second, while prion protein and α-synuclein fibrils need more than 50–100 mM NaCl present in solution to reach a plateau in solution OD_600_ values. This may be indicative of distinct fibrils with specific surface charges [44] that need to be shielded before effective association occurs. There also seems to be a correlation between when fibrils self-associate and the increase in ThT fluorescence intensity. This could be due to the same electrostatic repulsive forces acting upon both ThT molecules and other fibrils in solution.

Another interesting aspect is the massive effect ionic strength has on ThT binding and fluorescence properties. It is evident from this data that even minor variances in NaCl concentration result in major shifts of both fluorescence intensity, as well as the concentration of fibril-bound ThT molecules. This factor is important on two different fronts, both positive and negative. The negative aspect is that it makes comparisons of ThT fluorescence values, obtained at even slightly different ionic strength conditions, virtually impossible. Considering there are countless distinct conditions used to study amyloid formation and inhibition, a direct comparison between ThT fluorescence data sets becomes almost meaningless. The positive aspect, however, is the massive rise in ThT binding and fluorescence intensity upon the increase in ionic strength. If a certain type of fibrillar aggregate is difficult to track or detect due to its low dye affinity, one could achieve a several-fold rise in signal intensity by changing the solution’s ionic strength. In addition, ionic strength itself does not seem to alter the spectral properties of ThT, but rather causes it to induce a self-quenching effect due to increased binding.

In two of the cases, namely insulin and lysozyme fibrils, there exists a cut-off point, where the fluorescence intensity reaches the highest point and then begins to decrease. This is most likely caused by ThT fluorescence self-quenching, as the concentration of bound dye molecules surpasses a certain value, where quenching effects dominate over fluorescence. As this maximum intensity position is different for all four tested fibrils, this indicates that each type of amyloid aggregate has certain ideal total ThT concentration and solution ionic strength for monitoring ThT fluorescence.

The EEM position shift observed in all four cases is also an indicator that ionic strength may modulate ThT binding modes on the fibril’s surface. This may be due to certain positions with a maximum binding capacity, which causes the dye to associate with other, lower affinity positions. It could also directly affect the positions by altering the structure of the aggregate, as seen in the case of α-synuclein, or by increasing their affinity toward ThT by shielding charges between the fibril’s surface and dye molecules. In either case, the bound ThT distribution on the fibril varies with the solution’s ionic strength for all four types of aggregates examined in this work and such an effect may extend toward other amyloid aggregates as well. This ionic strength’s effect on dye binding may also explain why some anti-amyloid compounds are extremely effective under certain conditions, yet fail to produce any meaningful results under others [45]. If affinity between drug molecules and amyloid fibrils is the driving force of aggregation-inhibition and we assume that the effect of ionic strength extends beyond just fibril-fibril or fibril-ThT interactions, then it may also be directly responsible for the potential of inhibition.

Finally, it seems that ionic strength is not only an important factor which modulates ThT binding, but it can also affect the secondary structure of certain types of aggregates. While insulin, lysozyme and prion protein fibrils were not affected by changes in the solution’s ionic strength, α-synuclein fibrils displayed different amounts of disordered and beta-sheet structures at 0 and 2 M NaCl. Such an ionic strength-related structural shift is a clear indicator that the fibril environment is a determining factor in not just their formation, but their existence as well.

Taking everything into account, it appears that ionic strength is a highly important factor in fibril self-association events, structural aspects, ThT binding affinity and dye fluorescence intensity. It has a similar effect on distinct protein fibrils, including insulin, lysozyme, prion protein and α-synuclein amyloid aggregates. The significant dependence between ionic strength and fibril-dye interactions has both positive aspects, such as the ability to greatly enhance signal intensity, as well as negative ones—the diminished ability to compare fluorescence values between different conditions.

## 4. Materials and Methods

### 4.1. Fibril Formation

Human recombinant insulin powder (Sigma-Aldrich, St. Louis, MO, USA, cat. No. 91077C) was dissolved in a 100 mM sodium phosphate (pH 2.4) buffer, containing 100 mM NaCl. The final protein concentration was 200 µM (M = 5808 Da, ε_280_ = 6335 M*^−^*^1^cm*^−^*^1^). The solution was distributed to 1.5 mL test tubes (1.0 mL solution each) and incubated at 60 °C without agitation for 24 h. Afterwards, the test tubes were centrifuged at 10,000× *g* for 30 min and resuspended into MilliQ H_2_O. This centrifugation/resuspension procedure was repeated 4 times. After the final centrifugation step, the fibrils were resuspended into a smaller volume of H_2_O to result in a solution containing 400 µM fibrils.

Hen egg-white lysozyme powder (Sigma-Aldrich cat. No. L6876) was dissolved in a 50 mM sodium phosphate buffer (pH 6.0) containing 2 M guanidine hydrochloride (GuHCl) to a final protein concentration of 200 µM (M = 14,313 Da, ε_280_ = 37,970 M*^−^*^1^cm*^−^*^1^). The resulting solution was distributed to 1.5 mL test tubes (1.0 mL solution each, each containing two 3 mm glass-beads) and incubated at 60 °C for 72 h under 600 rpm agitation. Afterwards, the test tubes were centrifuged and resuspended into H_2_O as described in the insulin fibril preparation section.

Mouse prion protein 89–230 (MoPrP 89–230) was purified as described previously [38] without the His-tag cleavage step and was stored at −80 °C prior to use. The protein solution was mixed with 50 mM sodium phosphate (pH 6.0) buffers, which contained 0 or 6 M of GuHCl to a final protein concentration of 0.5 mg/mL and a GuHCl concentration of 0.5 M. The resulting solution was distributed to 2.0 mL test tubes (1.5 mL solution each) and incubated at 60 °C for 72 h under 600 rpm agitation. Afterwards, the test tubes were centrifuged and resuspended into H_2_O as described in the insulin fibril preparation section and the protein concentration was set to 400 µM (M = 18,621 Da, ε_280_ = 27,515 M*^−^*^1^cm*^−^*^1^).

α-synuclein was purified as described previously [46] and was stored at −80 °C prior to use. The protein solution was mixed with 10× phosphate buffer saline (PBS) and MilliQ H_2_O to a final protein concentration of 200 µM (PBS stock solution was diluted 10 times). The resulting solution was distributed to 1.5 mL test tubes (1.0 mL solution each, each containing 3 mm glass-beads) and incubated at 60 °C for 24 h under 600 rpm agitation. Afterwards, the test tubes were centrifuged and resuspended into H_2_O as described in the insulin fibril preparation section and the final protein concentration was set to 400 µM (M = 14,460 Da, ε_280_ = 5,960 M*^−^*^1^cm^−1^).In all four cases, prior to further use, the fibril solutions of each protein were combined, sonicated for 1 min using a Bandelin Sonopuls (Berlin, Germany) ultrasonic homogenizer equipped with a MS73 tip at 40% total power in order to avoid variation between each sample. This procedure was done in case one or more of the tested proteins can form a heterogenous mixture of fibrils. Afterwards, 1 mL aliquots were taken and sonicated for an additional 10 min with 30 s sonication/rest intervals to fragment and homogenize the fibrils in solution [47].

### 4.2. ThT and NaCl Solution Preparation

ThT powder (Sigma-Aldrich cat. no. T3516) was dissolved in MilliQ H_2_O to a concentration of 10 mM. The exact dye concentration was determined by diluting the solution 100 times and scanning the absorbance at 412 nm using a Shimadzu (Kyoto, Japan) UV-1800 spectrophotometer (ε_412_ = 23,250 M*^−^*^1^cm*^−^*^1^). The dye solution was then further diluted using H_2_O to a range of ThT concentrations, before mixing with fibril solutions.

NaCl was dissolved in MilliQ H_2_O to a final concentration of 4 M. The resulting solution was diluted to a range of NaCl concentrations using H_2_O, before mixing with fibril solutions.

### 4.3. ThT Fluorescence Assay

Fibril solutions were mixed with ThT and NaCl stock solutions in a 1:1:2 ratio to result in a final fibril concentration of 100 µM and a range of NaCl (0–2 M) and ThT (0–100 µM) concentrations. Each sample’s excitation/emission matrix (EEM) was scanned using a Varian Cary Eclipse spectrofluorometer (Agilent Technologies, Santa Clara, CA, USA) (excitation wavelength range was 435–465 nm with 1 nm steps, emission wavelength range was 460–500 nm with 1 nm steps). Excitation and emission slit widths were 5 nm and 2.5 nm respectively for insulin, lysozyme, and prion protein fibril samples and 2.5 nm*/*2.5 nm for α-synuclein samples. For each sample, three EEMs were recorded, averaged and a control EEM (without ThT) was subtracted. For an accurate comparison, the α-synuclein sample maximum fluorescence values were multiplied by a factor of 5, which is the intensity difference between scans when the excitation slit width is set to 5 nm or 2.5 nm.

### 4.4. Sample Absorbance and Optical Density Measurements

Each sample’s absorbance spectrum was scanned in the range from 300 nm to 600 nm using a Shimadzu UV-1800 spectrophotometer. For each case, three absorbance spectra were recorded, averaged and a control (MilliQ H_2_O) spectrum was subtracted. Afterwards, all samples were centrifuged at 10,000× *g* for 30 min, subsequently, a small aliquot (50 µL) of the supernatant was carefully removed (in order to not collect any pelleted fibrils) and diluted to 150 µL. The resulting sample absorbance spectra were also scanned as described previously. Optical density was measured at 600 nm using a 10 mm pathlength cuvette.

### 4.5. Inner Filter Correction and EEM Maxima Position Calculation

To account for the primary and secondary inner filter effect on fluorescence, caused by the sample’s absorbance, each EEM was corrected as described previously [48] using the following equation: I_m_ = I_c_ × 10^−(AEx+AEm)*/*2^(1)
where AEx is the sample’s absorbance at the excitation wavelength, AEm is the sample’s absorbance at the emission wavelength, I_m_ is the signal intensity observed during measurement and I_c_ is the corrected signal intensity.

The exact EEM intensity maxima position cannot be accurately determined due to signal noise. For this reason, a signal intensity “center of mass” was calculated by selecting the top 10% intensity values and using the following equation:λ = (∑ (λ_n_×∑I_n_))*/*∑ I_a_(2)
where λ is the wavelength of either the excitation or emission center of mass, λ_n_ is an excitation or emission wavelength, ∑I_n_ is the sum of all signal intensities at λ_n_, ∑ I_a_ is the sum of all signal intensities.

To negate the effect of Rayleigh scattering on the “center of mass” position, the region located 8 nm or closer to the excitation wavelength was not taken into account in the calculation.

### 4.6. Fourier-Transform Infrared Spectroscopy

Fibril samples were centrifuged at 10,000× g for 30 min and resuspended into D_2_O. This procedure was repeated 4 times. After the final centrifugation, the samples were resuspended into 200 µL D_2_O and divided into two equal volume parts. The divided samples were then mixed with either 100 µL D_2_O or D_2_O containing 4 M NaCl. Sample FTIR spectra were recorded as described previously [45]. A D_2_O spectrum was subtracted from the sample spectra, which was then baseline-corrected and normalized in the 1595–1700 cm^−1^ range. All data analysis was performed using GRAMS software.

## Figures and Tables

**Figure 1 ijms-21-08916-f001:**
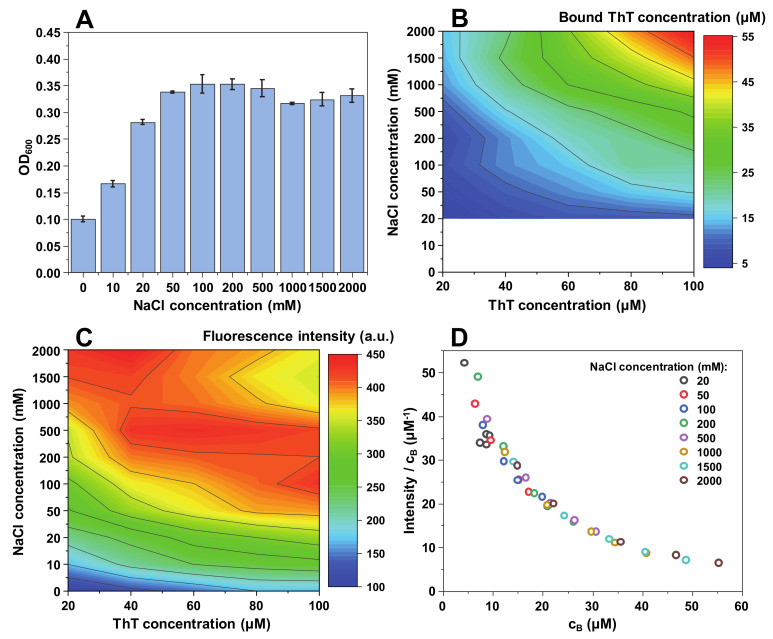
Insulin fibril and ThT solution optical density at 600 nm (**A**), bound ThT concentration (**B**), fluorescence intensity (**C**) and intensity/bound ThT (I/c_B_) ratio (**D**) at different NaCl concentrations. Optical density and fluorescence spectra intensities are the result of three repeats. The white area in the bound ThT graph (**B**) represents conditions under which bound ThT concentration could not be determined accurately. Fluorescence intensity values are corrected for the primary and secondary inner filter effects.

**Figure 2 ijms-21-08916-f002:**
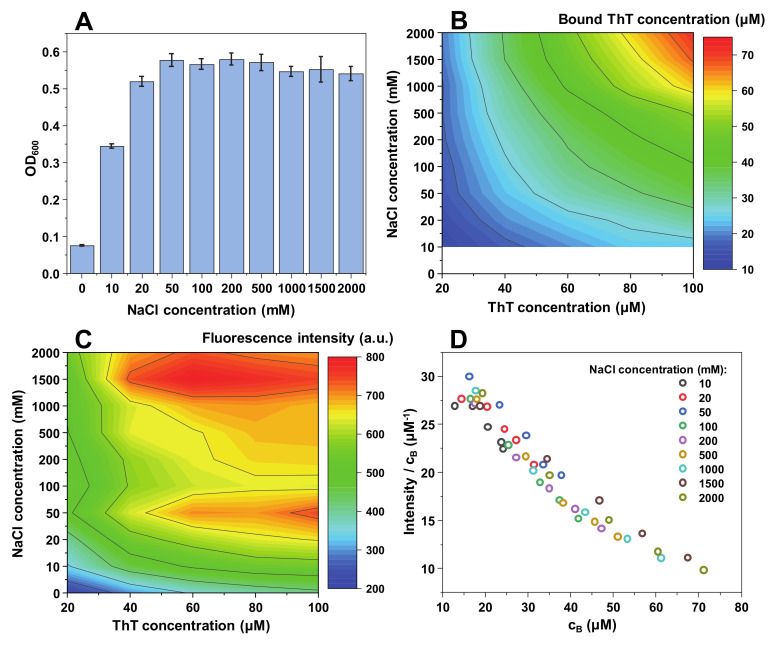
Lysozyme fibril and ThT solution optical density at 600 nm (**A**), bound ThT concentration (**B**), intensity/bound ThT (I/c_B_) ratio (**C**) and fluorescence intensity (**D**) at different NaCl concentrations. Optical density and fluorescence spectra intensities are the result of three repeats. The white area in the bound ThT graph (**B**) represents conditions under which bound ThT concentration could not be determined accurately. Fluorescence intensity values are corrected for the primary and secondary inner filter effects.

**Figure 3 ijms-21-08916-f003:**
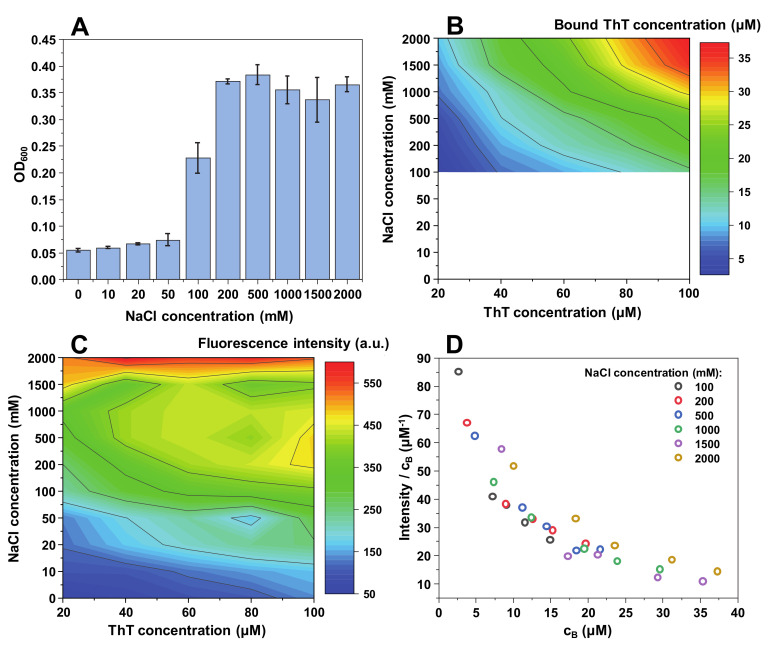
MoPrP fibril and ThT solution optical density at 600 nm (**A**), bound ThT concentration (**B**), intensity/bound ThT (I/c_B_) ratio (**C**) and fluorescence intensity (**D**) at different NaCl concentrations. Optical density and fluorescence spectra intensities are the result of three repeats. The white area in the bound ThT graph (**B**) represents conditions under which bound ThT concentration could not be determined accurately. Fluorescence intensity values are corrected for the primary and secondary inner filter effects.

**Figure 4 ijms-21-08916-f004:**
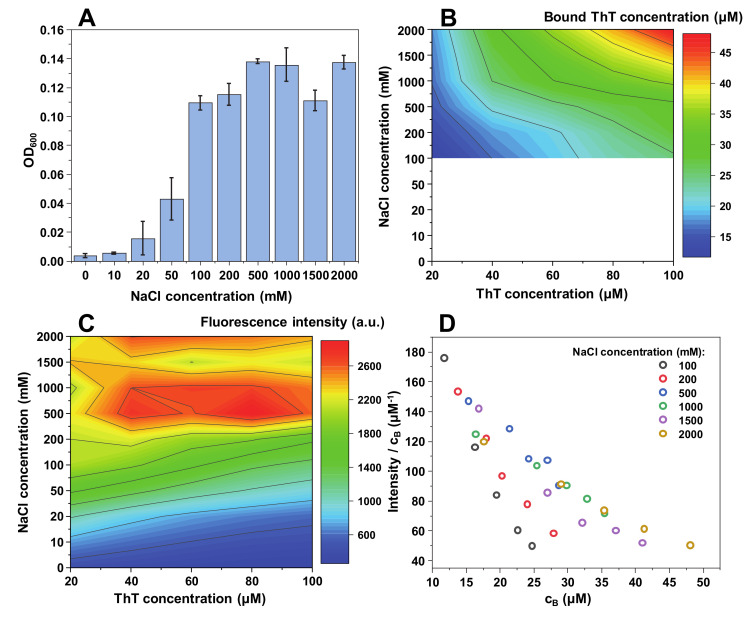
α-synuclein fibril and ThT solution optical density at 600 nm (**A**), bound ThT concentration (**B**), intensity/bound ThT (I/c_B_) ratio (**C**) and fluorescence intensity (**D**) at different NaCl concentrations. Optical density and fluorescence spectra intensities are the result of three repeats. The white area in the bound ThT graph (**B**) represents conditions under which bound ThT concentration could not be determined accurately. Fluorescence intensity values are corrected for the primary and secondary inner filter effects.

**Figure 5 ijms-21-08916-f005:**
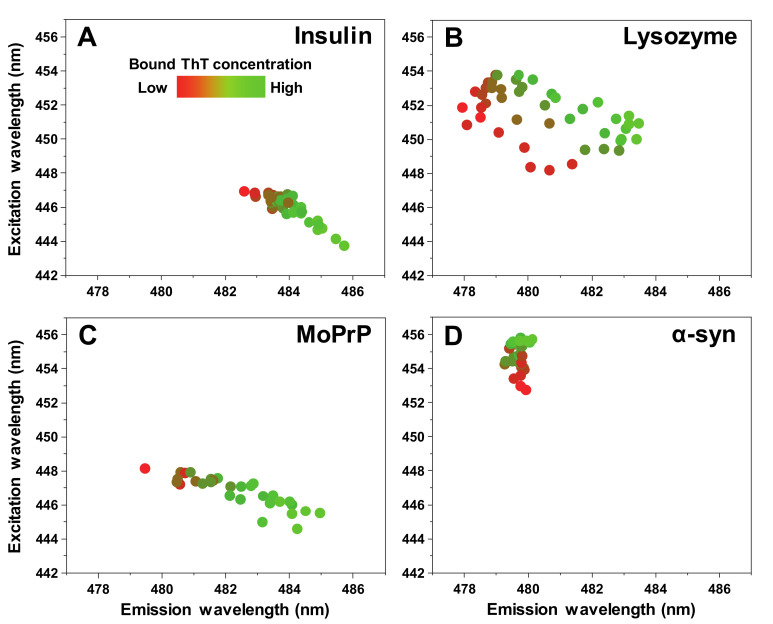
Insulin (**A**), lysozyme (**B**), mouse prion protein (**C**) and α-synuclein (**D**) fibril sample ThT fluorescence EEM intensity “center of mass” positions at different bound ThT concentrations.

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
