# Peer review of "Effect of Ionic Strength on Thioflavin-T Affinity to Amyloid Fibrils and Its Fluorescence Intensity"

_ijms, 2020, doi:10.3390/ijms21238916_

Round 1

Reviewer 1 Report

The mechanism of amyloid formation is still one of the top unsolved key question in biochemistry. Despite huge efforts to understand the mechanisms leading to severe diseases such as Parkinson's and Alzheimer's, and elucidate how the proteins stack together to form these lethal amyloid fibers, its mechanism is still completely unknown. In vitro, the formation of amyloid fiber kinetics can be monitored by the fluorescence increase of Thioflavin T, or ThT, which binds to the fibers and display elevated intensity when incorporated into secondary beta-rich structures. However, interpretation of this data is tricky as the fluorescence intensity can be modulated by other factors than the fibrillation kinetics itself. 

In this manuscript, Mikalauskaite et al. have performed a study to assess factors interfering with ThT fluorescence, using a model system consisting of insulin, lysozyme and mouse prion fibrils. Furthermore, the study also includes the effect of salt and ion strength and links its effects with ThT affinity and fluorescence emission upon fiber binding. 

Overall, I think the study is important since it highlights a usually forgotten problem, namely - how reliable are the ThT measurements that are conducted almost in all cases when monitoring amyloid structures. However, I think there are some comments below that need to be addressed before the manuscript can be published:

1.) Two of the most commonly studied amyloids are the Ab peptide and α-synuclein. It would have been interesting to see an additional example of either of these in the paper, especially α-synuclein which also is a highly charged protein. What were the authors rationale for not including at least one of these in the paper?

2.) Line 80 in the manuscript: The free and bound ThT should not be impossible to quantify. A proton NMR spectrum should be able to tell the amount of free ThT in solution when compared to a free ThT reference spectrum. I can agree however that it would not have added significant information to the paper, but it would have been possible to do.

3.) Why did the authors use OD600 as a measure for self-association instead of DLS for example? It would have been interesting to see DLS spectra at various NaCl concentrations as it would also have provided extra information of how these proteins behave in high salt concentrations. This would also detect sub-populations of potential oligomers in solution. I would like to see DLS experiments in the paper. I think it will strengthen the paper.  

3.) It is obvious from the study that NaCl makes a big difference on the ThT binding. How did the authors know that this effect is entirely due to the salt and not do different fiber shapes with various ThT affinity? Indirectly it would of course be an effect of the salt, but I think discussing this more is valuable.

4.) Even though the correlation between high NaCl concentrations and bound ThT is interesting, using a concentration of more than 200 mM NaCl in an experiment is not very common. Usually these experiments are conducted in PBS with a concentration of 140 mM NaCl. Everything higher are not psychically relevant anyways. What was the rationale for using these very high NaCl values?

Minor comment:

5.) In figure 4, is is quite hard to distinguish between the black and the dark green color. I would recommend changing the green to a red color to better see the data.

Author Response

See file attached.

Reviewer 2 Report

Some comments on the manuscript:

  • Pg 2 lines 51-52: it has been recently shown that ThT can bind to amyloidogenic proteins also in their native state (A covalent homodimer probing early oligomers along amyloid aggregation. Halabelian L, Relini A, Barbiroli A, Penco A, Bolognesi M, Ricagno S. Sci Rep. 2015 Sep 30;5:14651. doi: 10.1038/srep14651.)
  • Pg 2 line 74 amend “the first notable thing” “the first relevant observation”
  • Pg 3 line 102 and elsewhere: “absorbance at 600 nm”. Given that it is a suspension of insoluble aggregates I would rather speak of optical density rather than an absorbance that is typically recorded using clear solutions.
  • The Authors are expert of FTIR on amyloid fibrils. Did they check if the transfer of fibrils into different solutions with increasing NaCl concentrations does (or does not) trigger any conformational change to the fibrils which may account for different binding under different conditions?

Author Response

See file attached.

Round 2

Reviewer 1 Report

I have now read the revised manuscript from Mikalauskaite et al., and in my opinion the manuscript has been significantly improved by the introduction of FTIR data, and also by including another amyloid system for evaluation. I understand the authors' problem with the DLS experiments, as they mention that it might be hard to get reliable DLS data with too inhomogeneous samples. However, the FTIR spectra should be a good alternative method for evaluating the amyloid samples. 

Taken together, I am happy with the response from the authors and in particular due to the fact that they took the time to add α-synuclein into the paper as suggested. I have no further comments to the authors and I acknowledge the paper for publication.